# Thrombocytosis and abnormal liver enzymes: A trigger for investigation of underlying malignancy

Lucy C. Gold[1]*, Iain Macpherson[2◎], Jennifer H. Nobes[3◎], Eleanor Dow[3], Elizabeth Furrie[3], Scott Jamieson[4], John F. Dillon[2]

1 School of Medicine, Ninewells Hospital & Medical School, University of Dundee, Dundee, United Kingdom, 2 Division of Clinical and Molecular Medicine, Ninewells Hospital & Medical School, University of Dundee, Dundee, United Kingdom, 3 Department of Blood Sciences, Ninewells Hospital & Medical School, NHS Tayside, Dundee, United Kingdom, 4 Kirriemuir Medical Practice, Kirriemuir, United Kingdom

◎ These authors contributed equally to this work.
* lcgold@dundee.ac.uk

**Data Availability Statement:** All relevant data are within the paper and its Supporting Information files.

## Abstract

### Background

Thrombocytosis is often an incidental finding in primary care with a range of causes. Despite evidence of a strong association between thrombocytosis and malignancy, guidelines for investigating thrombocytosis in the absence of red flag symptoms remain unclear. A novel automated system of laboratory analysis, intelligent Liver Function Testing (iLFT), launched in Tayside in 2018 and has identified a patient group with thrombocytosis and abnormal liver test (LFT) results. This study analysed the outcome of these patients and investigated the use of thrombocytosis combined with LFTs in predicting risk of cancer.

### Methods and findings

Between August 2018 and August 2020, 6792 patients underwent iLFT, with 246 found to have both thrombocytosis and at least one abnormal LFT. A random case-matched control group of 492 iLFT patients with normal platelet count and at least one abnormal LFT was created. 7.7% (95% CI 4.7–11.8%) of patients with thrombocytosis had cancer compared to 2.0% (1.0–3.7%) of controls. Patients <40 years or with pre-existing causes of thrombocytosis were then excluded. Subsequent analysis revealed a 10.8% (6.6–16.3%) incidence of cancer in thrombocytosis patients (n = 176) compared to 2.5% (1.2–4.6%, p = 0.00014) in patients with normal platelet count (PLT) (n = 398). When thrombocytosis is combined with elevated alkaline phosphatase (ALP), there is a positive predictive value (PPV) of 20% for cancer. These rules were subsequently applied to a validation cohort of 71,652 patients, of whom 458 had thrombocytosis and elevated ALP. There was a 30.6% cancer incidence, confirming the strong predictive value of the combined test of PLT and ALP.

### Conclusions

These findings suggest a substantial increased risk of cancer in patients with thrombocytosis and raised ALP. This could be developed as an adjunct to current investigation

**Funding:** First author, LCG, received financial support from University of Dundee while undertaking the study as part of a Student Vacation Scholarship. The funders had no role in study design, data collection and analysis, decision to publish, or preparation of the manuscript.

**Competing interests:** I have read the journal's policy and the authors of this manuscript have the following competing interests: Eleanor Dow: Funding – Educational grants from Siemens and Abbott unrelated to this paper; Scott Jamieson: Declaration of interest – Reviewer of Scottish Referral Guidelines for Suspected Cancer. This does not alter our adherence to PLOS ONE policies on sharing data and materials.

algorithms, highlighting high-risk patients and prompting further investigation (such as computed tomography scans) where indicated.

## Introduction

Thrombocytosis is the term given to the presence of an elevated platelet count and is most commonly defined as a platelet count of above 450 x10$^9$/L [1]. It is often an incidental laboratory finding that can create a diagnostic predicament. Primary thrombocytosis is a common clinical finding in patients with myeloproliferative disorders such as essential thrombocythemia. These are uncommon disorders that arise as a result of genetic defects in platelet production [2].

Secondary, or reactive, thrombocytosis is a consequence of an underlying reactive or inflammatory process and is by far the most common cause of thrombocytosis amongst general populations. The most common clinical scenarios in which reactive thrombocytosis occurs include bacterial infections including tuberculosis; inflammatory disease such as rheumatoid arthritis or inflammatory bowel disease; malignant disease states; haemolytic anaemia; acute blood loss and peri-operative tissue damage [3]. The pathophysiology of reactive thrombocytosis is based around the upregulated production of thrombopoietin (TPO), the key regulatory hormone of platelet production [4]. Many interleukins, particularly interleukin-6 (IL-6), are often elevated in the acute-phase of neoplastic and inflammatory diseases and are known to upregulate the production of TPO messenger RNA (mRNA) in the liver, evidently causing a state of reactive thrombocytosis [5].

The clinical observation that thrombocytosis can occur in patients with solid organ malignancy was made over 100 years ago [6]. However, knowledge on the topic has remained stagnant for the majority of this time with an unclear relevance within clinical practice. Albeit, recent research has concluded that a finding of thrombocytosis in primary care conveys a significantly increased risk of cancer compared to those with normal platelet count [7]. On the back of this emerging evidence, Scottish national guidance for suspected cancer has recently incorporated thrombocytosis as a trigger for the investigation of lung, endometrial, oesophago-gastric and colorectal cancers when found alongside localising symptoms [8]. However, guidelines set by the Scottish Primary Care and Cancer Group for the management of a raised platelet count alone with no suspicion of a particular cancer are left vague and open to clinical interpretation. This lack of clarity creates the potential for an incidental finding of thrombocytosis to be overlooked despite its association with malignancy. Cancer is a leading cause for mortality in the UK with over 166,000 deaths in 2017 [9]. As early diagnosis of cancer has been recognised as a key strategy for improving cancer survival, any possible approach to improve early cancer recognition in primary care should be explored [10].

The growing burden and increasing mortality of liver disease across the globe has spurred the development of an automated system of liver function analysis known as the intelligent Liver Function Testing (iLFT) pathway. The algorithm incorporates liver enzymes with an array of aetiology screening tests and patient demographics to produce a detailed diagnosis and management plan for general practitioners (GPs) with the aim to improve early diagnosis of liver disease [11]. iLFT was made available across NHS Tayside in August 2018. When iLFT is requested, a series of further blood tests cascade automatically if one (or more) of the initial liver blood tests are abnormal. The cascade testing includes platelet count, allowing for the creation of a large cohort of patients with both abnormal liver enzymes and platelet anomalies.

These patients were investigated retrospectively with the intention of improving understanding around the significance of a thrombocytosis finding in primary care. In addition, the automatic incorporation of other blood markers within iLFT provided an opportunity to explore thrombocytosis in combination with abnormal liver enzymes to assess its effect on disease risk.

The aim of this study was to investigate the occurrence of malignancy in patients with thrombocytosis and explore the potential use of liver function tests in conjunction with a raised platelet count to develop a diagnostic tool for the prediction of patients at high risk of underlying cancer.

## Method

### Main cohort

Between August 2018 and August 2020, iLFT was requested in a total of 6792 patients by GPs in NHS Tayside. Amongst this cohort, 251 patients were noted to have a platelet count of $>400$ x$10^9$/L (above local laboratory reference range). Of the 251 thrombocytosis patients, 5 were excluded as insufficient clinical details were provided to allow the iLFT cascade to take place. This brought the total number in the thrombocytosis group to 246 patients.

To create a 2:1 matched case-control study, a random control group was selected from the remaining population of iLFT patients with a platelet count within the local laboratory reference range (150–400 x$10^9$/L). Two parameters, age and sex, were used to generate two matched controls for each patient of the study group creating a control group of 492 patients. The group contained patients with various liver enzyme abnormalities that triggered their iLFT cascade. A random number generator was used to carry out the selection and only the agreed parameters were used to distinguish between patient groups [12].

Exclusion criteria were established to eliminate patients with known causes of thrombocytosis as well as those at low risk of malignancy due to age (Table 1).

### Validation cohort

NHS Tayside laboratory results from a six-month period in 2016–2017 were retrieved and analysed to form a validation cohort. 71,652 patients were found to have both liver function tests and full blood count requested simultaneously in primary care. This cohort was then refined to only patients with thrombocytosis and specific abnormal liver enzymes (Table 1). Cancer incidence was then analysed following the same protocol as the main cohort.

**Table 1. Eligibility and exclusion criteria of main cohort including study and control group.**

| Eligibility Criteria | Exclusion Criteria |
|---|---|
| MAIN COHORT<br>iLFT request between August 2018 and August 2020<br>• Study = Thrombocytosis ($>400$x$10^9$/L)<br>• Control = Normal platelet count ($\leq$400 x$10^9$/L)<br>• Minimum of one abnormal liver function test result (alanine aminotransferase, bilirubin, alkaline phosphatase, gamma-glutamyl transferase)<br>VALIDATION COHORT<br>GP laboratory request for LFTs and FBC between August 2016 and February 2017<br>• Platelet count $>400$x$10^9$/L<br>• Alkaline phosphatase $>130$IU/L | $\leq$40 years old<br>Chronic inflammatory condition known prior to request<br>• Inflammatory bowel disease (diagnosed by gastroenterologist)<br>• Inflammatory arthropathy (diagnosed by rheumatologist)<br>Myeloproliferative neoplasm known prior to request (diagnosed by haematologist)<br>• Essential thrombocythemia<br>• Primary myelofibrosis<br>• Polycythaemia vera |

## Data collection

Local electronic records were used to retrieve information regarding the outcome of all patients. Imaging results and clinical communication letters were analysed to confirm any solid organ cancer diagnosis made up to 12 months from the initial laboratory request. A cancer diagnosis was counted if patients had confirmed histological or radiological evidence or if their specialist clinician had a strong clinical suspicion of underlying cancer. Any cancer diagnoses made after 12 months were not recorded in the results.

Patients with cancer known prior to iLFT were not recorded as 'cancer diagnoses' in the main cohort as to assess the use of iLFT as a diagnostic tool for cancer detection. Analysis of the validation cohort accepted cancer diagnoses if already present at time of request as it was not possible to only assess results that first detected the abnormality and therefore all cancers were included in order to evaluate the accuracy of the proposed test.

C-reactive protein (CRP) is often used clinically to confirm reactive thrombocytosis and was therefore investigated in the main cohort of patients. However, CRP is not always automatically incorporated in the iLFT cascade and therefore only patients who had CRP requested within 14 days of the iLFT request were recorded for analysis.

## Data analysis

All statistical analysis was carried out using IBM SPSS Statistics Version 25.0. The binomial test alongside the Clopper-Pearson interval determined cancer incidence and 95% confidence intervals for each cohort before and after exclusions. A Mann-Whitney U test was implemented to identify statistical significance. A p-value $<0.05$ was predetermined as statistically significant.

NHS Tayside Caldicott approval was obtained for the collection and retrospective analysis of patient data. The reporting of this study has been carried out as per the Strengthening The Reporting of Observational Studies in Epidemiology (STROBE) guidelines [13].

## Results

Initial analysis of the entire cohort revealed a 7.7% (95% Confidence Intervals (CI) of 4.7–11.8%) incidence of solid organ cancer in patients with thrombocytosis compared to 2.0% (1.0–3.7%) in patients with a platelet count within local reference range (150-400x$10^9$/L). Exclusion criteria was then applied to the thrombocytosis (Th) group and control (Co) group (Fig 1). 10.8% of the Th group were found to have a diagnosis of cancer within 12 months of iLFT analysis (95% CI 6.6–16.3%) compared to the Co group with a 2.5% incidence of cancer (1.2–4.6%), p = 0.00014 (Fig 2). This represents an approximate four-fold increased risk of cancer diagnosis within 12 months of a thrombocytosis result. In context, this relates to one in nine patients with an abnormal LFT result and unexplained thrombocytosis being diagnosed with cancer compared to a one in 40 among those with normal platelet count.

Liver enzymes incorporated within iLFT were then assessed to establish any correlation between abnormal LFTs and cancer diagnosis in conjunction with raised platelet count. After analysis of albumin, alkaline phosphatase (ALP), bilirubin, alanine transaminase (ALT), aspartate transaminase (AST) and gamma-glutamyl transferase (GGT), ALP was the only liver marker found to be of any relevance. The findings in relation to ALP and cancer diagnosis can be seen in Table 2. A clear relationship was found between ALP and cancer; this was enhanced in combination with PLT with a positive predictive value (PPV) of 20% for cancer detection. Compared to thrombocytosis alone, this combined with raised ALP almost doubles the risk of cancer diagnosis to one in five individuals. However, two of the 19 cancers in thrombocytosis patients were lost due to the addition of ALP.

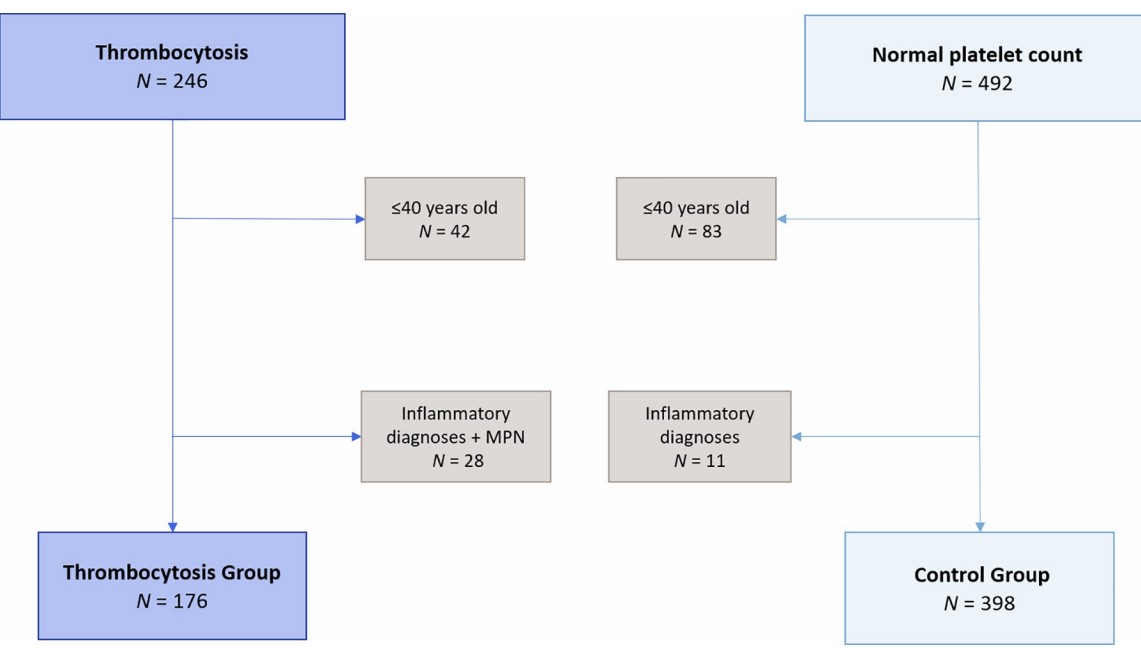

**Fig 1. Patient flow diagram of exclusion criteria being applied to thrombocytosis group and control group.** MPN, myeloproliferative neoplasm.

The use of CRP was then considered in the detection of high-risk cancer patients which can be seen in Table 2. However, as CRP is not routinely incorporated within the iLFT cascade, a large proportion of the patients were excluded. While the association between CRP and cancer diagnosis was similar to that of ALP, the reduction in patient numbers decreased reliability of the findings.

## Validation of findings

In order to validate the proposed diagnostic rule for cancer detection in patients with thrombocytosis and raised ALP, a validation cohort was determined, and the rule tested. After identification of 71,652 consecutive patients, the PLT, ALP and exclusion criteria were applied and the PPV for cancer was found to be 30.6% (95% CI 26.4–35.0%). This relates to one in three individuals having a solid organ cancer after the application of all proposed criteria (Fig 3).

CRP was also analysed amongst the validation cohort in those patients who had CRP ordered simultaneously to FBC and LFTs. Amidst 180 eligible patients with raised PLT, ALP and CRP, 54 cancers were detected, giving a PPV of 30.0%. Although this patient sample was lower in number, the evidence suggests CRP brought no additional value to the prospective diagnostic test.

## Platelet count distribution

Although all patients with thrombocytosis in the main and validation cohorts were known to have a platelet count of $>400x10^9$/L, the exact numbers were evaluated to assess any significance in their distribution (Fig 4). A large proportion of the patients were found to have a platelet count of $<450x10^9$/L, just above the local reference range of 150-400x10$^9$/L. No correlation was found between stage of disease and platelet count distribution.

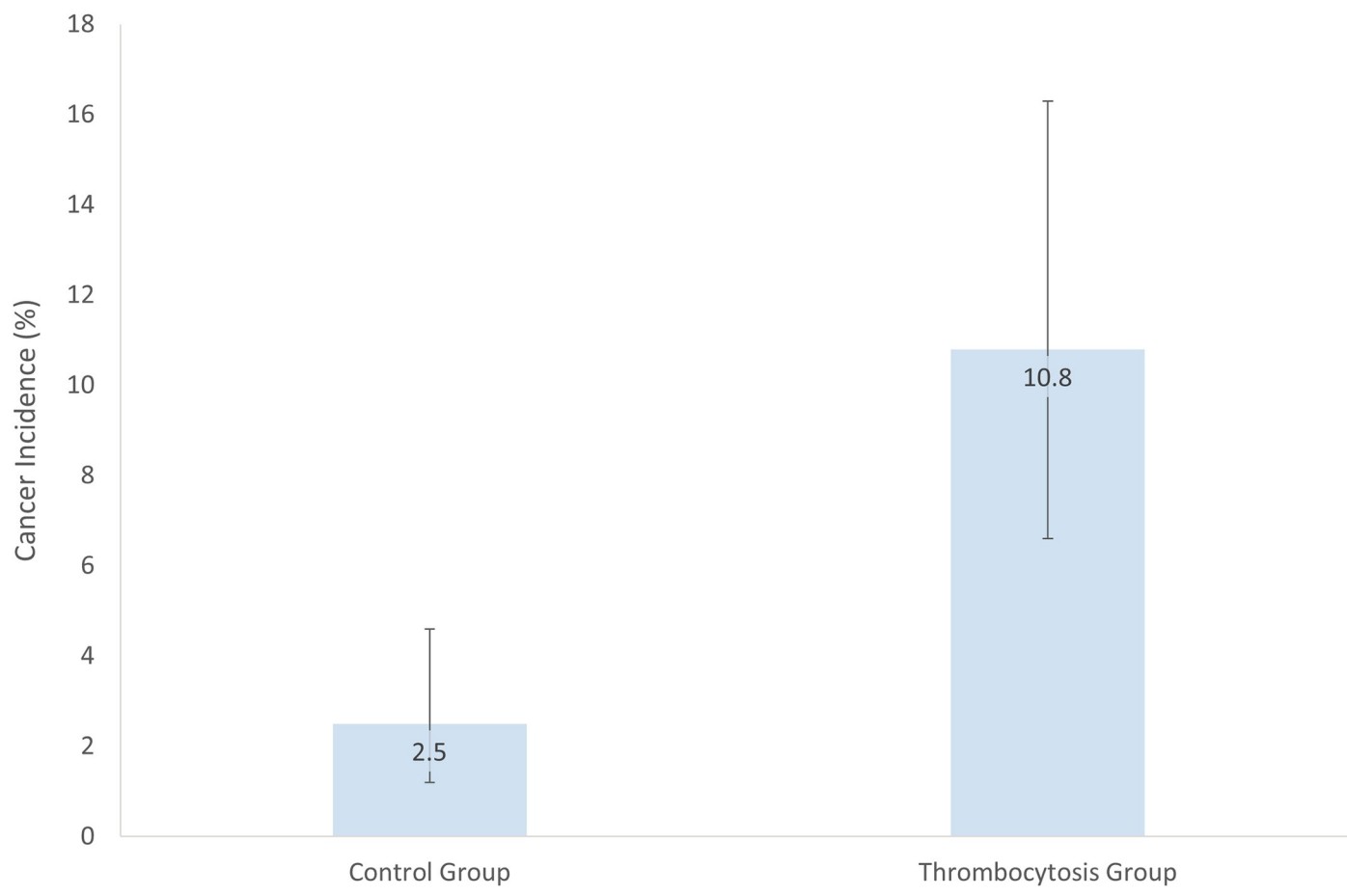

**Fig 2. Chart demonstrating cancer incidence in control group and thrombocytosis group with 95% confidence intervals shown.**

## Stage and site of disease

Finally, cancer site and stage of disease at time of diagnosis was investigated amongst both cohorts which is summarised in Table 3. Early or localised cancer made up a third of those diagnosed, however, metastatic colorectal cancer was found to make up the greatest proportion of cancers. This differs in comparison to national cancer statistics where colorectal adenocarcinoma is the fourth most common malignancy in men and women combined [14]. Liver metastases were also seen in a large number of patients with advanced disease, as were bony and lung metastases. Despite these findings, the study was not powered to detect strong

**Table 2. Reliability and validity of all proposed tests for cancer detection.**

| Test | N of +ve patients (cancers/total) | PPV | NPV | Sensitivity | Specificity |
|---|---|---|---|---|---|
| PLTC | 19/173 | 11% | 97% | 66% | 72% |
| ALP | 27/212 | 13% | 99% | 93% | 66% |
| PLT + ALP | 17/83 | 20% | 98% | 59% | 88% |
| CRP | 19/120 | 16% | 97% | 83% | 59% |
| PLT + CRP | 12/64 | 19% | 95% | 52% | 79% |

PLTC, platelet count; ALP, alkaline phosphatase; CRP, c-reactive protein; PPV, positive predictive value; NPV, negative predictive value.

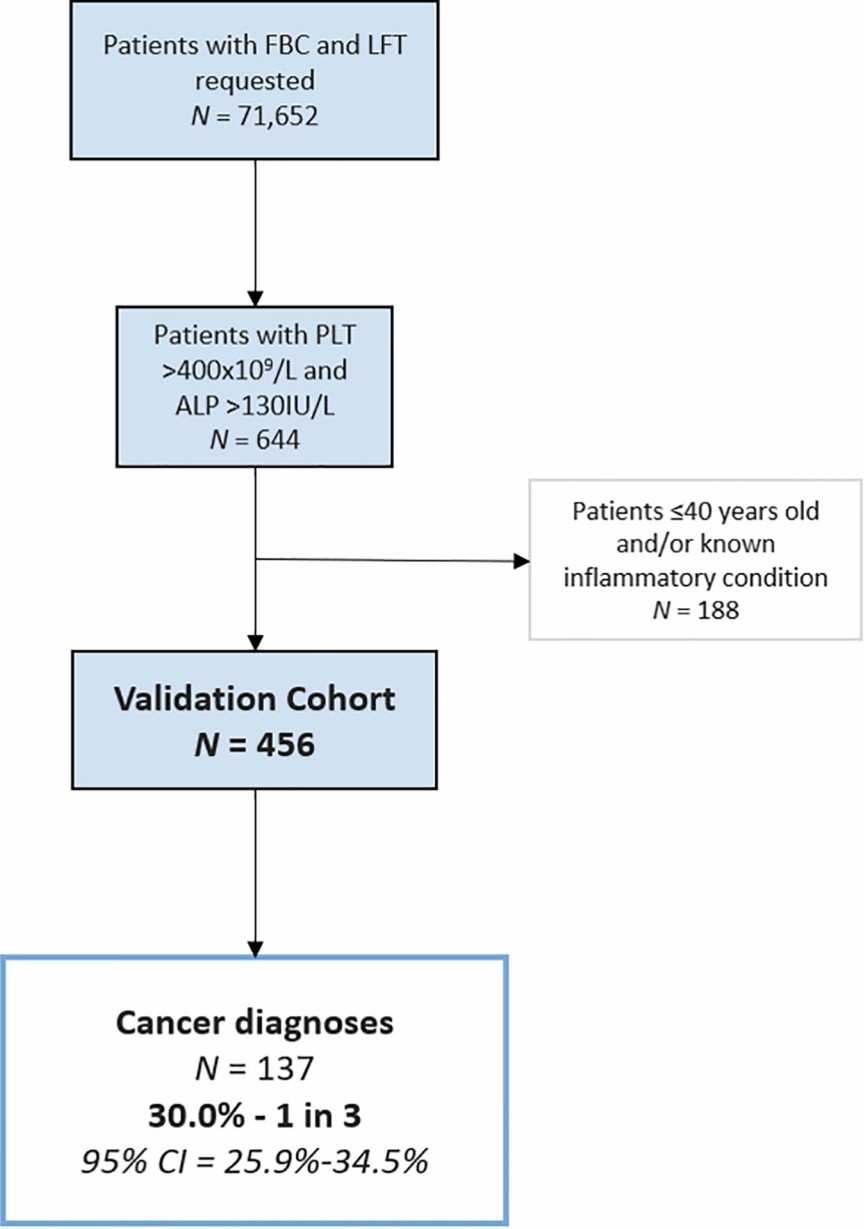

**Fig 3. Flowchart depiction of formation of validation cohort with exclusion criteria applied.** Number of cancers found within cohort shown with 95% confidence intervals (CI).

associations between particular cancers and thrombocytosis. The large scale of previous studies were able to confirm the high predictive value of thrombocytosis in lung and colorectal cancer [7, 15].

## Discussion

### Main findings

This retrospective cohort study has confirmed the significance of a thrombocytosis finding in primary care with a 10.8% incidence of solid organ cancer compared to 2.5% in patients with

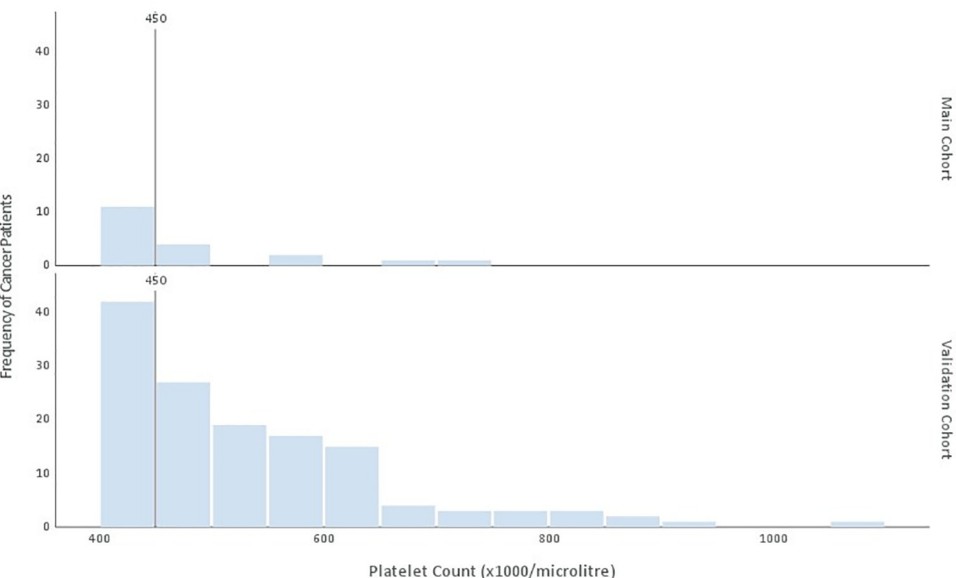

**Fig 4. Histogram displaying distribution of platelet count in cancer patients in main and validation cohorts.**
Vertical line represents 450x10$^9$, the upper limit of the reference range for platelet count in many laboratories.

platelet count within local reference range (150-400x10$^9$/L). These significant findings led to the development of a simple yet robust set of diagnostic criteria for the detection of patients at highest risk of underlying malignancy. After the exclusion of patients under 40 years old and those with existing diagnoses of inflammatory or myeloproliferative conditions, it was concluded that patients with thrombocytosis and raised alkaline phosphatase have a 20% cancer risk. This proposed 'rule' was validated using a cohort of 71,652 patients which confirmed and strengthened this prediction with a positive predictive value of 30.6% or one in three patients presenting with this combination of results.

The sensitivity of the PLT/ALP rule for the detection of cancer is modest at 59%, however this does not prevent it from having some clinical benefit. The association of thrombocytosis with malignancy is well known, but equally recognised to be of weak predictive value. The use of this diagnostic rule has the potential to trigger further investigation by putting a significant numeric risk to the clinical observation using analytes that are routinely measured and therefore maintaining low opportunity cost. Although the rule cannot be expected to detect all underlying cancers, the significant number it does detect allows for its use as a decision support tool for general practitioners, especially if part of an automated investigation and advice system such as iLFT.

## Comparison to existing literature

In review of existing literature, the association between thrombocytosis and cancer detection in primary care has only been thoroughly investigated in recent years. A prospective study was carried out by Bailey *et al.* (2017) to investigate overall risk of cancer with thrombocytosis using a UK wide database [7]. The study of over 30,000 patients found that of patients with thrombocytosis, 11.6% of male patients and 6.2% of females were diagnosed with cancer within one year of the result, compared to a control group of which 4.1% and 2.2% of men and women had cancer, respectively. The results of the present study are comparable to the findings of Bailey *et al.* as they both confirm an overall increased risk of cancer with thrombocytosis and support the implementation of further investigation in these patients.

**Table 3. Cancer stage, site and site of metastasis amongst all cancer patients in both main and validation cohorts combined.**

| Disease Stage | | | Disease Site | | |
|---|---|---|---|---|---|
| | *N* | % of total | | *N* | % of stage total |
| Local | 58 | 37.2 | Lung | 13 | 22.4 |
| | | | Oesophageal | 5 | 8.6 |
| | | | Prostate | 5 | 8.6 |
| | | | Pancreas | 4 | 6.9 |
| | | | Colorectal | 3 | 5.2 |
| | | | Breast | 3 | 5.2 |
| | | | Ovarian | 3 | 5.2 |
| | | | Renal | 3 | 5.2 |
| | | | Mixed organ | 3 | 5.2 |
| | | | Other* | 16 | 27.6 |
| Advanced | 98 | 62.8 | Colorectal | 30 | 30.6 |
| | | | Prostate | 14 | 14.3 |
| | | | Lung | 12 | 12.2 |
| | | | Renal | 9 | 9.2 |
| | | | Pancreas | 6 | 6.1 |
| | | | Breast | 5 | 5.1 |
| | | | Gallbladder | 4 | 4.1 |
| | | | Mixed organ | 3 | 3.1 |
| | | | Other* | 15 | 15.3 |
| Metastasis Site | | | Liver | 55 | |
| | | | Bone | 27 | |
| | | | Lung | 23 | |
| | | | Brain | 9 | |
| | | | Peritoneal | 6 | |
| | | | Nodal | 5 | |
| | | | Pancreas | 1 | |
| | | | Adrenal | 1 | |

*all cancers with ≤2 patients.

Additionally, a recent systematic review by the same author analysed features of individual cancers in primary care and found that some, but not all, cancers had thrombocytosis as an early marker of disease and no biological or anatomical link between cancers was identified [15]. Study analysis from Bailey *et al* (2017) found the odds ratio of colorectal cancer amongst thrombocytosis patients to be insignificant in published multivariable models. Despite this, there was an increase in bowel cancer incidence in patients with thrombocytosis compared to the general population within this study. The greater proportion of bowel cancer diagnoses may be a result of the particular patient cohort and the circumstances in which they attended their GP. In addition, the most common location for bowel cancer metastases is the liver which could account for both thrombocytosis and abnormal liver enzymes [16]. Ultimately, an increase in detection of bowel cancer amongst the cohort of patients is unsurprising. However, the considerable proportion of lung cancers, both local and advanced, support the findings of Bailey *et al.* which named lung cancer within the most strongly associated cancers after a thrombocytosis finding. In addition to lung cancer, oesophago-gastric and uterine cancers were found to have strong associations with raised platelet count, this was not reflected in this study, however, this may be a result of the addition of ALP in the validation cohort.

ALP is known to be a non-specific marker of disease and is therefore not commonly recognised as a diagnostic tool for cancer. However, it has previously been concluded that raised ALP alongside other pathological entities should point towards suspicion of cancer [17]. ALP had also been found to be a useful prognostic indicator in colorectal cancer by Saif *et al.* (2005), with increased levels of ALP correlating with increased stage of disease, as well as being associated with liver metastases [18]. Over half of the cancer patients with thrombocytosis did have liver metastases at the time of diagnosis. In addition, bone metastases are known to cause a substantial increase in ALP; this was exemplified by the highest ALP readings being taken from numerous patients with skeletal metastases from primary prostate cancer. Overall, ALP alone is not suitable as a diagnostic marker for cancer due to its lack of sensitivity and specificity, however, it clearly has a strong link with advanced malignancy and therefore combining ALP with thrombocytosis creates an effective case-finding tool.

## Strengths and limitations

The size of the main cohort was reasonable with 738 patients analysed from an initial sample group of 6792 patients. However, the number of identified cancers detected in the Th group was low with only 19 cases detected. Nevertheless, the validation cohort was retrieved from a data set of over 70,000 patients which successfully confirmed the conclusions of the main cohort and provided a far greater number of cancer patients for analysis.

An additional limitation was the retrospective nature of the study; this was somewhat rectified by the application of a randomly selected case matched control group to reduce the possibility of bias. The control group was made up of patients who had also attended their GP and required blood testing as this was a more accurate control than the general healthy population. The recording of cancer diagnoses was completed manually with the potential for error and bias. However, criteria for all diagnoses were established to alleviate this and only patients with cancer diagnoses recorded in secondary care communications were included. The recent primary care records of patients would have been beneficial in excluding other causes of thrombocytosis such as acute infectious illness.

CRP was investigated with the possibility of refining the at-risk group and increasing the PPV for cancer detection. While both of these were achieved, sample numbers were dramatically reduced, decreasing the statistical power of the findings. As a result of this and the lack of additional benefit found in the validation cohort, CRP was not incorporated within the PLT/ALP rule, however, further research into its association with thrombocytosis and cancer would be worthwhile.

Furthermore, the study findings have provided a clear justification for further investigation of patients after a finding of thrombocytosis and raised ALP. However, the direct impact of the proposed rule on the morbidity and mortality of cancer patients has not been explored at this time. The potential benefit of earlier diagnosis is far more likely amongst patient with local disease, however the impact on patients with metastatic disease is unclear and requires further investigation.

## Implication for clinical practice

**Platelet count reference range.** In addition to the diagnostic rule, another interesting finding with relevant implications for clinical practice is the distribution of platelet count amongst patients with solid organ cancer. The possibility exists that practitioners may have a tendency to ignore borderline abnormal results with the assumption that they are insignificant [19]. Interestingly, a large proportion of said patients had a platelet count between 400-450x$10^9$/L which is just above local reference range. The substantial proportion of cancer

patients with only slight anomalies of platelet count highlights the clinical significance of thrombocytosis and cancer and the rationale for alerting GPs to the risk of disease.

**Method of further investigation.** The National Institute for Health and Care Excellence (NICE) threshold for the investigation of suspected cancer is a risk of >3% [20]. This is exceeded by a thrombocytosis finding alone and surpassed ten times over by the PLTC/ALP rule, making it suitable for use in clinical practice. The remaining question is how best to investigate patients found to be at high risk of underlying disease. As no cancer type in particular is suspected in these patients, a generalised investigation is necessary. A computed tomography (CT) scan is an appropriate medical imaging technique as they allow whole-body tumour detection. The use of CT scans for population-wide cancer screening has long been rejected due to cost-ineffectiveness and unnecessary radiation exposure [21]. However, there is a recognised potential benefit for its use in a well-defined group of patients such as high-risk potential lung cancer patients [22]. With a cancer detection rate of 20–30% in patients that meet all criteria of the proposed rule, a CT scan is justifiable.

## Conclusion

A thrombocytosis finding should raise suspicion of cancer after the emerging evidence in recent years, however, the lack of definitive national guidance results in many cases not being investigated. The proposed PLT/ALP rule has the potential to alert GPs of the significant risk surrounding a patient with thrombocytosis and abnormal liver enzymes and give guidance on appropriate next steps to ensure it is not overlooked. UK cancer survival is lagging behind other European countries and therefore every effort should be made to enhance early diagnosis as a crucial strategy for improvement [23]. Primary care is at the forefront of early diagnosis and the PLT/ALP rule provides a means of detecting patients with no red flag symptoms and a 20–30% risk of underlying malignancy. These study findings have shed some light on the clinical problem of unexplained thrombocytosis and encourage the implementation of the PLT/ALP rule into primary care practice.

## Supporting information

**S1 Data.**
(XLSX)

## Acknowledgments

Ian Kennedy, Blood Sciences IT, Department of Blood Sciences, NHS Tayside, Ninewells Hospital & Medical School, Dundee, UK.

## Author Contributions

**Conceptualization:** Eleanor Dow, Elizabeth Furrie, Scott Jamieson, John F. Dillon.

**Data curation:** Lucy C. Gold, Jennifer H. Nobes.

**Formal analysis:** Lucy C. Gold.

**Investigation:** Lucy C. Gold.

**Methodology:** Iain Macpherson, Jennifer H. Nobes, Eleanor Dow, Elizabeth Furrie, John F. Dillon.

**Supervision:** Iain Macpherson, John F. Dillon.

**Writing – original draft:** Lucy C. Gold.

**Writing – review & editing:** Lucy C. Gold, Iain Macpherson, Jennifer H. Nobes, Scott Jamieson, John F. Dillon.

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
