## [Decision Letter · Decision Letter 0]

20 Dec 2021

PONE-D-21-26649Thrombocytosis and Abnormal Liver Enzymes: A Trigger for Investigation of Underlying MalignancyPLOS ONE

Dear Dr. Gold,

Thank you for submitting your manuscript to PLOS ONE. After careful consideration, we feel that it has merit but does not fully meet PLOS ONE’s publication criteria as it currently stands. Therefore, we invite you to submit a revised version of the manuscript that addresses the points raised during the review process.

We look forward to receiving your revised manuscript.

Kind regards,

Hsu-Heng Yen

Academic Editor

PLOS ONE

Journal Requirements:

2. In ethics statement in the manuscript and in the online submission form, please provide additional information about the patient records/samples used in your retrospective study. Specifically, please ensure that you have discussed whether all data/samples were fully anonymized before you accessed them and/or whether the IRB or ethics committee waived the requirement for informed consent. If patients provided informed written consent to have data/samples from their medical records used in research, please include this information.

3. You indicated that ethical approval was not necessary for your study. We understand that the framework for ethical oversight requirements for studies of this type may differ depending on the setting and we would appreciate some further clarification regarding your research. Could you please provide further details on why your study is exempt from the need for approval and confirmation from your institutional review board or research ethics committee (e.g., in the form of a letter or email correspondence) that ethics review was not necessary for this study? Please include a copy of the correspondence as an "Other" file.

No

I have read the journal's policy and the authors of this manuscript have the following competing interests: Eleanor Dow: Funding - Educational grants from Siemens and Abbott; Scott Jamieson: Declaration of interest - Reviewer of Scottish Referral Guidelines for Suspected Cancer

We note that one or more of the authors are employed by a commercial company: Siemens and Abbott

Reviewers' comments:

Reviewer's Responses to Questions

**Comments to the Author**

1. Is the manuscript technically sound, and do the data support the conclusions?

Reviewer #1: Yes

Reviewer #2: Yes

2. Has the statistical analysis been performed appropriately and rigorously? 

Reviewer #1: Yes

Reviewer #2: I Don't Know

3. Have the authors made all data underlying the findings in their manuscript fully available?

Reviewer #1: Yes

Reviewer #2: No

4. Is the manuscript presented in an intelligible fashion and written in standard English?

Reviewer #1: Yes

Reviewer #2: Yes

5. Review Comments to the Author

Reviewer #1: This is a nice study that attempts to detect patients with cancer by a simple lab analysis and that could be very helpful in GP practice. Although I consider the findings very encouraging they should be validate prospectively and probably redefining limits of platelets to gain more sensitivity.

I would like to make two simple comments;

“Analysis of the validation cohort accepted cancer diagnoses if already present at time of request”. It may me think if this would be a bias and I would like a comment about it.

Was there a trend in the number of platelets in patients with cancer? Do patients with advanced cancer had more platelets? Maybe, due to the retrospective analysis there is data about it.

Reviewer #2: Nice paper that highlights the utility of combination of readily available data to detect clinically important outcomes.

It would be worth clarifying if the control group had the same liver enzyme abnormality as the thrombocytosis group or whether this was a mixed bag of enzyme abnormalities.

The cut off of platelet count of 400 is an arbitrary reference range level for the laboratory. With so many values being in the 400 to 450 range, were other values explored such as 350 or 375?

On line 198 this refers to figure 3 not figure 2

On line 208 this refers to figure 4 not figure 3

Table 2 could be rearranged in order site of metastases is below rather than alongside the site of the advanced cancers.

6. PLOS authors have the option to publish the peer review history of their article (what does this mean?). If published, this will include your full peer review and any attached files.

Reviewer #1: No

Reviewer #2: No

---

## [Author Response · Author response to Decision Letter 0]

31 Jan 2022

Please see Response to Reviewers

---

## [Decision Letter · Decision Letter 1]

17 Mar 2022

PONE-D-21-26649R1Thrombocytosis and Abnormal Liver Enzymes: A Trigger for Investigation of Underlying MalignancyPLOS ONE

Dear Dr. Gold,

Thank you for submitting your manuscript to PLOS ONE. After careful consideration, we feel that it has merit but does not fully meet PLOS ONE’s publication criteria as it currently stands. Therefore, we invite you to submit a revised version of the manuscript that addresses the points raised during the review process.

We look forward to receiving your revised manuscript.

Kind regards,

Hsu-Heng Yen

Academic Editor

PLOS ONE

Journal Requirements:

Reviewers' comments:

Reviewer's Responses to Questions

**Comments to the Author**

1. If the authors have adequately addressed your comments raised in a previous round of review and you feel that this manuscript is now acceptable for publication, you may indicate that here to bypass the “Comments to the Author” section, enter your conflict of interest statement in the “Confidential to Editor” section, and submit your "Accept" recommendation.

Reviewer #1: All comments have been addressed

Reviewer #3: All comments have been addressed

2. Is the manuscript technically sound, and do the data support the conclusions?

Reviewer #1: Yes

Reviewer #3: Yes

3. Has the statistical analysis been performed appropriately and rigorously? 

Reviewer #1: Yes

Reviewer #3: Yes

4. Have the authors made all data underlying the findings in their manuscript fully available?

Reviewer #1: Yes

Reviewer #3: Yes

5. Is the manuscript presented in an intelligible fashion and written in standard English?

Reviewer #1: Yes

Reviewer #3: Yes

6. Review Comments to the Author

Reviewer #1: (No Response)

Reviewer #3: *The authors have come with an insightful work that highlights a trigger for investigation of underlying malignancy in patients with combined unexplained thrombocytosis and deranged liver enzymes (particularly ALP). The authors have tried to absorb the salient comments forwarded by the previous reviewers. In general, the manuscript is well written, and it should be considered for publication provided that the suggested modifications both from the reviewers and academic editor(s) are taken into account. Below are the main comments for the review report.

*Introduction: line ≠69: “Thrombocytosis is the term given to an overproduction of platelets…” Megakaryocyte proliferation is not the only cause of thrombocytosis as it can also result from decreased platelet sequestration (as in the case of asplenia, for example). Therefore, the sentence should be rewritten. One option of modification is: thrombocytosis is the presence of elevated platelet count… or the word ‘usually’ should be added to the existing sentence.

*Introduction: line ≠77 & 78: tuberculosis is written separately from bacterial causes as if it were not a bacterial infection. It should be modified like: The most common clinical scenarios in which reactive thrombocytosis occurs include bacterial infections including tuberculosis…

*Regarding exclusion criteria, was there any consideration of conditions that could potentially cause raised alkaline phosphatase such as growth healing fractures, osteomalacia, hyperparathyroidism, and hyperthyroidism? I would like to hear about this.

7. PLOS authors have the option to publish the peer review history of their article (what does this mean?). If published, this will include your full peer review and any attached files.

Reviewer #1: No

Reviewer #3: **Yes: **Subah Abderehim Yesuf

---

## [Author Response · Author response to Decision Letter 1]

28 Mar 2022

Please see Response to Reviewers

---

## [Editor Report · Decision Letter 2]

4 Apr 2022

Thrombocytosis and Abnormal Liver Enzymes: A Trigger for Investigation of Underlying Malignancy

PONE-D-21-26649R2

Dear Dr. Gold,

We’re pleased to inform you that your manuscript has been judged scientifically suitable for publication and will be formally accepted for publication once it meets all outstanding technical requirements.

Kind regards,

Hsu-Heng Yen

Academic Editor

PLOS ONE
---

## [Editor Report · Acceptance letter]

13 Apr 2022

PONE-D-21-26649R2 

Thrombocytosis and Abnormal Liver Enzymes: A Trigger for Investigation of Underlying Malignancy 

Dear Dr. Gold:

I'm pleased to inform you that your manuscript has been deemed suitable for publication in PLOS ONE. Congratulations! Your manuscript is now with our production department. 

Kind regards, 

on behalf of

Dr. Hsu-Heng Yen 

Academic Editor

PLOS ONE